# An Investigation into Indoor Radon Concentrations in Certified Passive House Homes

**DOI:** 10.3390/ijerph17114149

**Published:** 2020-06-10

**Authors:** Barry Mc Carron, Xianhai Meng, Shane Colclough

**Affiliations:** 1School of Natural and Built Environment, Faculty of Built Environment, Creative and Life Sciences, South West College, Enniskillen BT74 4EJ, UK; 2School of Natural and Build Environment, Faculty of Engineering and Physical Sciences, Queens University Belfast, Belfast BT7 1NN, UK; x.meng@qub.ac.uk; 3School of Architecture, Planning and Environmental Policy, Faculty of Engineering and Architecture, University College Dublin, D04 V1W8 Dublin, Ireland; shane.colclough@ucd.ie

**Keywords:** indoor radon, certified passive house, mechanical ventilation with heat recovery

## Abstract

The Energy Performance of Buildings Directive (EPBD) has introduced the concept of Nearly Zero Energy Buildings (NZEB) specifying that by 31 December 2020 all new buildings must meet the nearly zero- energy standard, the Passive House standard has emerged as a key enabler for the Nearly Zero Energy Building standard. The combination of Passive House with renewables represents a suitable solution to move to low/zero carbon. The hypothesis in this study is that a certified passive house building with high levels of airtightness with a balanced mechanical ventilation with heat recovery (MVHR) should yield lower indoor radon concentrations. This article presents results and analysis of measured radon levels in a total of 97 certified passive house dwellings using CR-39^3^ alpha track diffusion radon gas detectors. The results support the hypothesis that certified passive house buildings present lower radon levels. A striking observation to emerge from the data shows a difference in radon distribution between upstairs and downstairs when compared against regular housing. The study is a first for Ireland and the United Kingdom and it has relevance to a much wider context with the significant growth of the passive house standard globally.

## 1. Introduction

### 1.1. Background

The UK Parliament became the first in the world to declare a climate emergency on 1 May 2019. The Irish government declared a climate emergency a week later, and the Northern Ireland Assembly followed suit in February 2020 [1]. The Paris Agreement, signed in 2016, is an agreement within the United Nations Framework Convention on Climate Change (UNFCCC), dealing with greenhouse-gas-emissions mitigation, adaptation, and finance [2]. The most recent report issued by the Intergovernmental Panel on Climate Change (IPCC) is called the Special Report on Global Warming of 1.5 °C (SR15). The report assesses projected impacts at a global average warming of 1.5 °C and higher levels of warming. Its crux finding is that meeting a 1.5 °C target is possible but will require deep emissions reductions and rapid, far-reaching and unprecedented changes in all aspects of society [3]. The influential Emissions Gap Report advocates for the passive house standard in its 2016 edition, consolidating a recommendation for the standard as a climate mitigation solution in the IPCC 4th assessment report released in 2007 [4]. Both recommendations recognise that buildings are central to meeting the sustainability challenge, as currently European buildings account for approximately 40% of total energy consumption within the European Union (EU) [5].

The Energy Performance in Buildings Directive (EPBD) mandates that all EU member states build Near Zero Energy Buildings (NZEB) by 2021. The EPBD defines near zero energy buildings, in broad terms, as those with high levels of energy efficiency, with the very low amount of energy required to be provided to a very significant degree by energy from renewable sources, preferably produced on or near site [6]. The UK’s Climate Change Committee (CCC) is an independent, statutory body established under the Climate Change Act 2008 [7]; in 2019, they published a report titled “UK housing: Fit for the future?” [8] which states that; “Greenhouse gas emission reductions from UK housing have stalled, and homes across the UK must be improved now to address the challenges of climate change. The costs of building to a specification that achieves the aims set out in this report are not prohibitive and getting design right from the outset is vastly cheaper than forcing retrofit later”. The CCC is calling for a UK “ultra-energy efficient standard” with a space heating demand of 15–20 kWh/m^2^/yr. ideally heated with an electric Heat Pump [8].

The International Passive House Standard offers a proven methodology to achieve this standard. The combination of Passive House with heat pump and renewable energy production presents a cost-effective solution to move to the required NZEB standard. Passive Houses focus on energy saving and are designed to have an energy demand that is as low as practically achievable. With such a small amount of energy to be supplied, it is easier to meet the subsequent demand with renewable sources [9].

### 1.2. Motivation for this Study

However, while the passive house standard is being recognised as a potential significant contributor, there are areas of research which are key to health and which have not been addressed to date. Existing passive house related research in the UK and Ireland has predominantly focussed on Energy Consumption, Indoor Air Quality (IAQ) and overheating [10]. There is a small number of studies in this space [11,12,13]; however, no research in the UK or Ireland has investigated the relationship between the unique characteristics of certified Passive House buildings and indoor radon concentrations. This is underlined by the National Radon Control Strategy in Ireland, which identified a key knowledge gap: that “the relationship, if any, between increased air tightness and elevated radon levels is unknown”. This research study is aimed to provide evidence to help understand this knowledge gap. Given the impact radon concentration can have on health issues such as lung cancer risk, a focus is now needed in this important area.

### 1.3. Study Objectives

This study aims to test the hypothesis that certified Passive House buildings, with the associated high levels of air tightness coupled with mechanical ventilation, will result in a reduction in indoor radon gas concentrations compared to conventional buildings. It includes the following detailed objectives:(a)Evaluating the findings against the established reference levels and the national average;(b)Comparing radon distribution levels between upstairs and downstairs;(c)Identifying the influence of construction materials on corresponding radon concentrations;(d)Determining the indoor radon concentrations of the Passive House Retrofit standard (EnerPHit) sample;(e)Carrying out comparative case studies analysis of indoor radon concentrations in a high-risk radon area.

### 1.4. Review of Current Literature

Passive House (or Passivhaus) refers specifically to the International Passive House Standard as developed, defined and administered by the Passive House Institute (PHI) in Darmstadt, Germany. Passive House has a very clear set of requirements, so it is possible to check if a building meets the definition of the Passive House Standard. Rigorous modelling and verification are required in the design and construction stages to meet Passive House certification standards. This research will monitor only Passive House buildings certified by the PHI [14]. To meet the Passive House Standard, the airtightness of a building must achieve an air change per hour rate of less than 0.6 air changes at 50 Pascals of pressure (n50), and have ventilation provided by a balanced mechanical ventilation heat recovery (MVHR) system. This minimizes energy loss to the outside, improves insulation performance and reduces moisture ingress into the building fabric [15]. This approach contrasts sharply with natural ventilation methods where sufficient ventilation for occupants is often achieved, in part, due to a leaky building fabric. The resultant draughts in naturally ventilated buildings are often exacerbated using open fires which further draw in air for combustion. As concerns about IAQ and health grow, ensuring good Indoor Air Quality (IAQ) is critical. Available research already indicates that a correctly installed and operating MVHR system has a positive effect on IAQ and humidity levels [16]. The Passive House Standard uses the European air quality category IDA 3 (Moderate IAQ CO_2_ level 600–1000 ppm) to define MVHR operating parameters along with output that is based on the number of people (30 m^3^/h per person) according to Deutsches Institut fur Normung (DIN) 1946, the German standard for ventilation [17]. The Passive House certification criteria mandates key metrics for compliance in respect to MVHR including early design consideration, successful installation and commissioned units. This is confirmed by academic research into the performance of the Passive House Standard [16]. The standard is based on compliance of the International Standard for Thermal Comfort ISO 7730 [18].

The Passive house standard is the fastest growing energy performance building standard in the world [19]; this was the reason for selecting this sample for investigation. As outlined, the concept of reducing air infiltration through the envelope and utilising heat recovery with mechanical ventilation are inherent to the success of the standard. It is well known that ventilation in buildings has an influence on indoor radon concentrations [20]. Mechanical Ventilation systems can induce negative pressure or also exert positive pressure on the building envelope. With respect to radon negative pressure on the building, envelope has potential to accelerate the migration of radon into the dwelling [20]. Certified passive house buildings however pose less risk with balanced systems and thus reduced pressure differentials. In order to gain full certification, the MVHR system must be balanced within 10% and best practise is seen as <3% [17]. Finally, the ventilation design employs the crossflow principle, to ventilate circulation areas. This is where air moves from supply to extract points through the building, thus avoiding the need, usually, for supply air points in hallways, landings, stairs, etc. To do this air transfer between all ventilated rooms is essential (see Figure 1), and this is normally done by providing a 10 mm undercut beneath internal doors, although the exact amount can be calculated according to the designed air transfer rates [21]. Intuitively the floor plan configuration will also have an impact [22].

Radon is a naturally occurring, radioactive gas that results from the decay of uranium in rocks and soils. It is the major source of ionizing radiation exposure to the population. Radon decays to form tiny radioactive particles, some of which stay suspended in the air as colourless, odourless, tasteless gas that can only be measured using special equipment. Normally, when radon is emitted into the open air, it is quickly diluted to harmless concentrations. However, when radon enters an enclosed space, (such as a house) through cracks in floors or gaps around pipes and cables, it can build up to a dangerously high concentration. Inhaled radon particles give a radiation dose that may damage cells in the lung [23]. The World Health Organisation (WHO) has identified radon as a known human carcinogen and has reported a wealth of biological and epidemiological evidence connecting radon exposure and lung cancer [24]. Radon is estimated to cause 1100 deaths per year in the UK (and is the second largest identified cause of lung cancer after smoking) [25], and approximately 300 cases of lung cancer in Ireland every year can be linked to radon [26]. Considering that the typical person in industrial countries (such as the UK and Ireland) spends approximately 90% of their time indoors [27], there are surprisingly few academic studies on radon in the home.

Monitoring indoor radon is of fundamental importance, and this research represents an opportunity to advance an understanding of the effect of increasing energy performance standards and the role of increased airtightness and mechanical ventilation. Public Health England (PHE) in the UK estimates that with an increase in radon concentration of 100 Becquerels per cubic metre (Bq/m^3^), the risk of a smoker developing lung cancer increases by up to 31% with a central estimate of 16% [28]. Other researchers [29,30,31], however, hold a conflicting view. They argue that this probabilistic approach demonstrates that cohort and case-control epidemiologic studies cannot reveal the true shape of the dose–response relationship for radon-induced lung cancer. They conclude that there is no significant risk among low radon concentrations advocating the *“as low as reasonably achievable”* (ALARA) principle.

Radon is measured in Becquerels per cubic metre of air (Bq/m^3^). The WHO proposes a reference level of 100 Bq/m^3^ to minimize health hazards due to indoor radon exposure. However, if this level cannot be reached under the prevailing country-specific conditions, the chosen reference level should not exceed 300 Bq/m^3^, which represents approximately 10 mSv per year according to recent calculations by the International Commission on Radiation Protection [24]. The Reference Level for homes in Ireland is 200 Bq/m^3^. The Reference Level represents a radon concentration above which action to reduce radon levels is likely to be needed [24]. In the United Kingdom, Public Health England (PHE) issued advice (HPA, 2010) recommending that radon exposure should be reduced where the annual average radon concentration exceeds the Action Level (AL) of 200 Bq/m^3^ and ideally to below the Target Level (TL) of 100 Bq/m^3^ [32]. Therefore, in summary, in the context of the passive house standard having been identified and a potential significant contributor to reduced energy consumption in dwellings, this study focuses on the knowledge gap identified in the critical area of radon concentrations in housing, which has been certified as complying to the passive house standard. To do so it reports on measurements carried out on a cohort of dwellings in Ireland and the UK.

## 2. Materials and Methods

The sample in this study comprises of 97 certified Passive House buildings in Ireland and the UK and consists of two-house classifications, 92 are passive house certified and five meet the passive house EnerPHit standard (namely passive house retrofit). In addition to these, 25 comparison homes were also selected simply because of their proximity to corresponding certified homes.

The oldest of these passive house homes was built in 2005 with the most recent being constructed in 2019. The entire sample is also all two-story domestic dwellings. The largest of these is 455 m^2^ while the smallest is 104 m^2^. Of the 97 homes, 58 are of masonry construction, and the remaining 39 are of timber frame construction. All the homes are passive house certified, and all have a balanced mechanical ventilation, heat recovery unit and have an airtightness level (n50) of <0.6 for new homes and for the EnerPHit <1.0. The building characteristics and materials are significant, as the most common sources of radon are gas from the soil/ground and off-gassing from building materials containing radon [33]. Building material radon emissions are much lower than radon gas being emitted from soil/ground gas and only apply to such building materials as ground rock and those that originate from ground rock (e.g., sand, soil and cement). Concentrations of radon present in these building materials will vary, depending upon geological origin [34]. The small number of homes retrofitted to the EnerPHit standard are significant as other studies show that energy retrofitting of homes may reduce the potential for ventilation flushing of radon gas from the house, thereby increasing radon levels [35]. Retrofit houses may also have an existing floor that does not include radon protection, and sealing the full footprint of the building may consequently prove difficult. Therefore, it is difficult to predict the effect of applying Passive House techniques to existing buildings on indoor radon concentrations; a properly installed and operating MVHR system could reduce the radon level but failing to completely seal the building envelope could increase the radon level.

The methodology to test the hypothesis was to survey as many certified passive house buildings using the same standard and protocols as employed under the national radon strategy to facilitate a comparative framework. The method of measurement used to test indoor radon concentrations was by CR-39^3^ alpha track diffusion radon gas detectors placed in the main living area (Room 1) and the main bedroom (Room 2) for just over three months. This was carried out over three different monitoring phases from October 2017–June 2019. Radon results are presented as an annual average using the seasonal adjusted average (SAA) method. The test results are compared with the national reference levels, national averages and existing survey data on radon.

An independent assessment of measurement uncertainties of the radiological Protection institute of Ireland (RPII) radon measurement service was carried out by previously published [36]. The Radon laboratory holds the accreditation from the International Organisation for standardisation and International Electrotechnical Commission (ISO/IEC) 17025. A requirement of this standard is an estimate of uncertainty of measurement. This work employed two approaches to estimate the uncertainty. The bottom up approach involved identifying the components that were found to contribute to the uncertainty. Estimates were made for each of these components, which were combined to give a combined uncertainty of 13.5% at a Rn concentration of approximately 2500 Bq/m^3^ at the 68% confidence level. By applying a coverage factor of k = 2, the expanded uncertainty is ±27% at the 95% confidence level. The top down approach used information approximately 95% confidence level. This is good agreement for such independent estimates [36]. In this research, as the monitoring and testing was carried out by this lab, the uncertainty level of ± 27% was adopted for the study. The presentation of results incorporates this percentage in the error bars in graphs.

## 3. Results

Radon measurements were completed in a total of 122 homes, 97 certified (including 5 EnerPHit) passive house buildings and 25 comparison homes. None of the 97 certified passive homes surveyed had radon concentrations exceeding the 200 Bq/m^3^ national Reference Level (see Figure 2). Only 6.79% of the sample breached the target level or WHO recommendation of 100 Bq/m^3^. The maximum concentration measured was 166 Bq/m^3^ in a home in Northern Ireland, which was in a defined higher risk area.

The Environmental Protection Agency (EPA) carried out the National Radon Survey (NRS) of Ireland between 1992–1999 [26]. The survey characterized areas of Ireland in terms of their radon risk, and one of the key findings was that the geographic weighted national average indoor radon concentration at that time was 89 Bq/m^3^. Since then, several developments have taken place in Ireland that are likely to have impacted on the national average radon concentration.

Two of the most significant include the introduction of amended Building Regulations in 1998, requiring radon preventive measures in new buildings in High Radon Areas (HRAs), and the significant rate of new builds that grew by 47% between 1999–2014. To re-assess the national average indoor radon concentration, a survey protocol was carried out in 2015, to measure radon in a sample of homes representative of radon risk and geographical location. This new national average was published in 2017 and could then be used to assess the effectiveness of the measures that have impacted on this metric since it was first established in the 2002 NRS [26]. The results showed that the national average indoor radon concentration for homes in Ireland was 77 Bq/m^3^, a decrease from the 89 Bq/m^3^ reported in the 2002 NRS. This figure of 77 Bq/m^3^ is now a baseline metric for the National Radon Control Strategy (NRCS) [26].

### 3.1. Radon Levels against National Average

The average indoor radon level based on the 97 certified passive homes monitored in this study is 36 Bq/m^3^ as shown in Table 1. It can be directly compared with the national average of 77 Bq/m^3^. The radon level of certified passive homes is directly compared with the non-passive houses (namely comparison homes), which were also monitored in this study. The average of comparison homes was found to be 88 Bq/m^3^, which is broadly in line with NRS.

### 3.2. Radon Distribution

The single most striking observation to emerge from the data shows a difference in radon distribution between upstairs and downstairs when compared against regular housing. In previous UK research, radon levels were found to be typically 35% lower on first floor bedrooms compared to ground floor living rooms [33]. In the current research, the radon concentrations between both floors tested found that levels were only 6% lower on the first-floor bedrooms compared to ground floor living rooms. Therefore, with radon levels significantly lower in the passive house sample, it warranted further investigation. An analysis of 344 standard two-story homes from the EPA 2015 NRS and the Passive House sample of 97 two-story homes presented this different radon distribution. Distribution ratios shown in Table 2 of the bedroom/living room radon levels in the standard two-story homes presented a Gaussian distribution (mean 0.79, median 0.74 with a standard deviation of 0.37). The results of the certified passive house sample are significantly anomalous (mean 1.03, median 0.92 with a standard deviation of 0.56).

A possible reason for the difference may lie in the fact that the Passive house standard has a defined specification for airtightness and balanced MVHR systems, unlike much of the standard dwellings. In addition to this, there is a consistent framework on design, installation and commissioning of these systems. This quality assurance coupled with typical layout of a two-storey dwelling combine to produce the lower indoor concentrations and closer distribution levels between upstairs and downstairs. Figure 3 below illustrates the contrast in radon distribution between floors in a comparison on this passive house study and the National Radon Survey (NRS) 2015.

### 3.3. Radon from Building Materials

While the influence of building materials on indoor radon concentrations is recognized, there is a paucity of quantitative data representing the structural contribution to domestic radon. A figure of 20 Bq/m^3^ has recently been suggested for the contribution from building materials to indoor radon concentrations [33]. In this passive house study, 58 dwellings are constructed from masonry and the remining 39 dwellings are of timber frame construction. The analysis of the results shown in Figure 4 reveals that the timber frame sample has a slightly lower radon level than the masonry group. This corresponds with previous research [37]. However, as the number of houses investigated in this research was relatively small, and it was not the key focus of the research, the results should be treated with caution.

### 3.4. Comparative Case Studies

A total of 10 comparative case studies were also carried out, which were in known high risk areas. For each case study, the comparison between a certified passive house and the home directly next door was performed. It is noted that a significant variance can be found between neighbouring dwellings. The correlation of these findings is still interesting as the conventional homes demonstrate elevated radon levels in all 10 case studies. As shown below in Figure 5:

The figure above illustrates that in the five case studies with the clear differentiation, (5, 6, 8, 9 and 10), these conventional homes all have radon levels on or above the Action Level of 100 Bq/m^3^. Moreover, of note in Figure 5, Homes 8 and 9 have significantly elevated levels 598 Bq/m^3^ and 409 Bq/m^3^. By comparison, the corresponding passive houses next door in both case studies have levels of 67 Bq/m^3^ and 166 Bq/m^3^, respectively.

Figure 6 above presents a comparison of the EnerPHit or Passive House retrofit homes against the comparison homes measured in this study. The average result for the EnerPHit sample was 72 Bq/m^3^ which was lower than the standard house sample of 88 Bq/m^3^.

## 4. Discussion

This research focused on using the same common radon testing method employed for national surveys in both Ireland and the UK, in order to facilitate comparative analysis to examine the implications of passive house standard for indoor radon concentrations in the domestic construction. The findings from this study provide evidence that explain the influence of attributes of the passive house standard.

(a)The results of this study are broadly in agreement with the small amount of similar research carried out in other EU Countries. The Passive House sample presented an average 36 Bq/m^3^. This falls below all other comparable reference levels and is aligned with the *“as low as reasonably achievable”* (ALARA) principle. There are two likely causes for the reduced concentrations: The role of airtightness coupled with a properly installed and commissioned balanced mechanical ventilation system facilitate lower indoor radon concentrations. The role of passive house certification also provides level of oversight to ensure the design metrics are upheld with respect to airtightness and ventilation rates and balancing of the system.(b)The radon distribution results are interesting in that they presented with a much lower ratio between upstairs and downstairs. The observed correlation between upstairs and downstairs might be explained in this way: The passive house standard employs the crossflow principle for mechanical ventilation design which allows circulation and air transfer throughout the building thus redistributing radon concentrations. These results are in sharp contrast to the limited available research on radon distribution between floors. In the Passive House certification framework, the air transfer strategy is checked to ensure certification.(c)Another possible explanation for this more uniform distribution ratio is the role of floor plan configuration; under the ventilation design methodology in passive house, there will often be more extract than supply on the ground floor and indeed the opposite on the first floor. This is primarily because all bedrooms are supply and in a typical two storey dwelling the larger percentage of bedrooms are located upstairs. This suggestion will also contribute to a lower distribution ratio between floors.(d)The comparison of the construction materials did result in two differentials. The timber sample yielded a lower radon concentration of 32 Bq/m^3^ compared to 38 Bq/m^3^ in the masonry sample. There was also a difference in radon distribution both floors had an average of 29 Bq/m^3^ in the timber sample compared to 40 Bq/m^3^ on the ground floor and 36 Bq/m^3^ on the first floor in the masonry sample. It is reported in previous literature that masonry construction materials will emit and contribute to the background radon in a building. The results here would indicate the same; however, this would require further investigation.(e)The results from the EnerPHit standard sample indicated higher radon levels 72 Bq/m^3^ than the rest of the passive house sample 36 Bq/m^3^; however, this was still a lower level than the comparison sample measured in this study 88 Bq/m^3^ and the figures recorded in the last national radon survey 77 Bq/m^3^. This suggests that the passive house standard for retrofit will also act as an effective control for radon. It must be noted, however, that this must be treated with caution as the number of EnerPHit homes measured (five) is very low.

## 5. Conclusions

### 5.1. Summary of Research

This research has provided further evidence of the knowledge gap on the relationship between improved energy efficient design strategy’s and direct influence on indoor radon concentrations. The five main objectives have been realised, specifically:(a)The monitored radon levels have been taken on 97 certified passive house buildings, and these results have been evaluated against the established reference levels and the national average.(b)The radon distribution levels between upstairs and downstairs were analysed in in the 97 certified passive house buildings, which are all two storey domestic dwellings.(c)The type of construction was identified for all buildings in the sample, and the influence of the construction materials was compared.(d)Ten homes within the sample were identified as being in high risk radon areas and had the distinction of having an adjacent dwelling. The results of the monitoring of indoor radon concentrations were compared between both homes at each of the ten locations.(e)The sample of certified buildings included five Passive House Retrofit (EnerPHit) homes, and the radon levels for these homes were also compared.

### 5.2. Main Findings

The main findings of this study, based on the results of monitoring are as follows:(a)The overall monitoring results support the hypothesis outlined at the start of this paper that certified Passive House buildings perform better in respect to indoor radon concentrations 36 Bq/m^3^ compared to conventional homes 88 Bq/m^3^ given less airtightness and no MVHR systems. The case study dwellings located in the geogenic high-risk areas, which presented elevated concentrations in the standard homes compared directly with the ten passive house buildings, further consolidate the narrative.(b)The radon distribution results support the opinion that the passive house framework for quality assurance of the design, installation and commissioning delivers design performance in MVHR systems. This will result in more extract on the ground floor thus reducing the ground floor radon level and lowering the distribution gap.(c)The findings on the relationship between construction materials and radon concentrations are not statistically significant to make claims about background radon emissions.(d)Finally, the EnerPHit sample presents lower indoor radon concentrations. It is acknowledged that the number of houses investigated was very small, and the results should be treated with caution; however, they correlate with results in similar pilot studies.

### 5.3. Implications for Future Practise

This project is the first comprehensive investigation of indoor radon in certified passive house buildings in the UK and Ireland. The analysis of the radon concentrations has extended our knowledge of the influence of airtightness and mechanical ventilation heat recovery systems as delivered by the passive house standard. The findings of this study have the following important implications for future practice:(a)They illustrate the value of having a clear methodology of quality assurance (integral to the passive house process).(b)Passive House exhibits better radon performance, which may be ascribed to the combination of reduced air infiltration combined with MVHR.(c)They highlight the significance of ensuring the MVHR system is balanced at the commission stage, so the house is at a slight positive pressure, i.e., the slight positive pressure (<3%) will reduce the risk of radon ingress.

### 5.4. Future Work

Further work needs to be done to translate knowledge generated in this research.

(a)Modelling or simulation studies of the crossflow principle and the relationship with MVHR airflows.(b)Investigations into the radon concentrations emanating from construction materials particularly with the recent EU Directive mandating reference levels for gamma radiation emission from building materials.(c)In addition, more work could be carried out on the pressure differentials produced from the various arrays of mechanical ventilation systems being employed in European housing under the EBPD.(d)Finally, a replication of this work could be carried out on a larger sample of passive house EnerPHit projects, and this study is replicable and scalable for different regions with passive house growth such as North America and Canada where radon is a prevalent issue.

## Figures and Tables

**Figure 1 ijerph-17-04149-f001:**
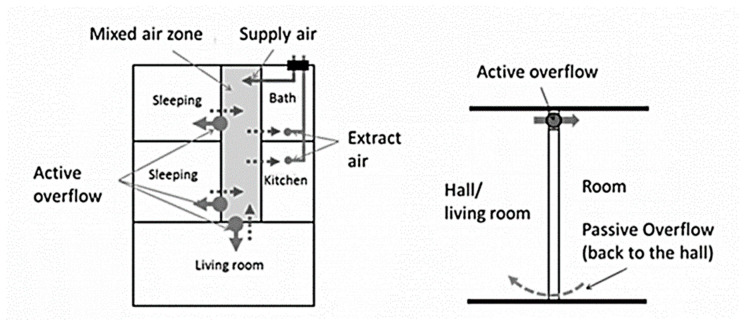
Passive House Ventilation concept with active overflow. Source: Passive House Institute (PHI) [19].

**Figure 2 ijerph-17-04149-f002:**
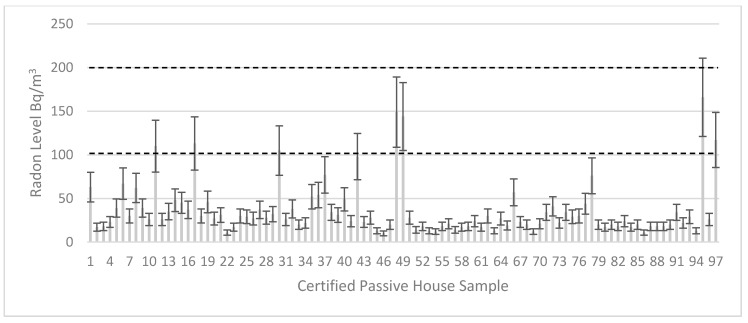
Passive House Sample (97 homes) with National Reference/Action Level 200 Bq/m^3^ and WHO recommendation/Target Level 100 Bq/m^3^ shown with the dashed lines.

**Figure 3 ijerph-17-04149-f003:**
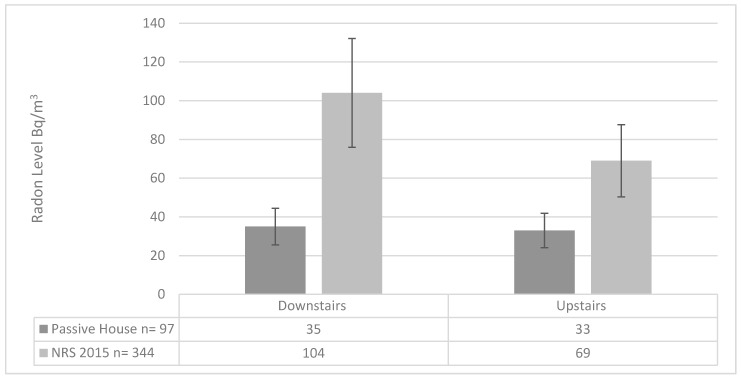
Radon Distribution comparison of data by floor between this study and the NRS 2015.

**Figure 4 ijerph-17-04149-f004:**
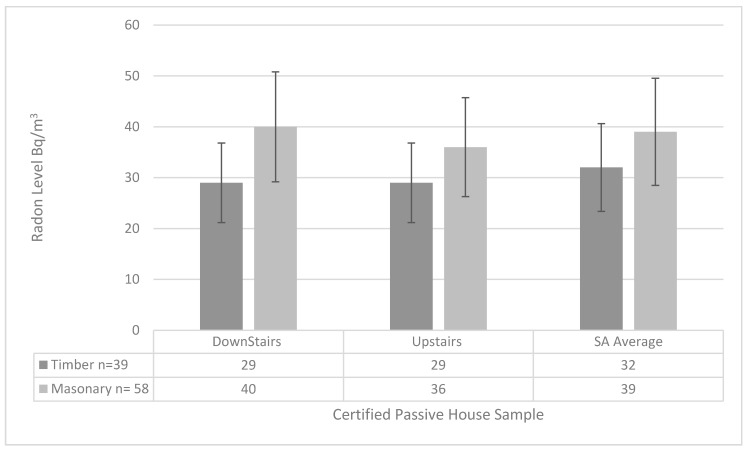
Radon monitoring construction materials.

**Figure 5 ijerph-17-04149-f005:**
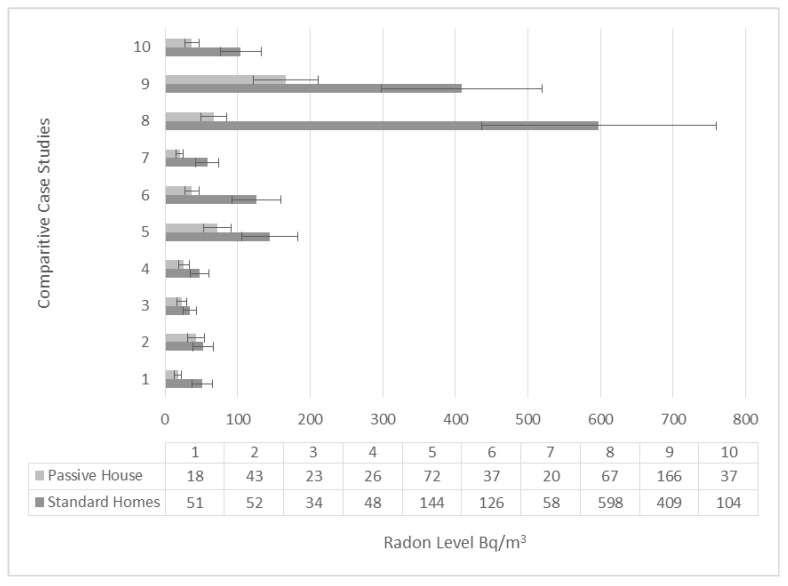
Direct comparative case studies.

**Figure 6 ijerph-17-04149-f006:**
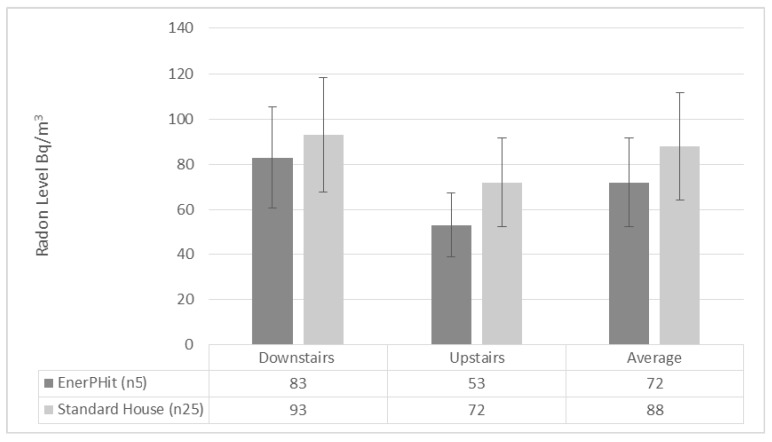
Passive House Retrofit standard (EnerPHit) comparison.

**Table 1 ijerph-17-04149-t001:** Radon results showing the EPA 2015 National Radon Survey (NRS), the comparison sample and finally the Passive House sample.

Metric	EPA 2015 NRS	Comparison Sample	PH Sample
Number of homes measured	649	25	97
No. of homes > 200 Bq/m^3^	8%	8%	0%
No. of homes > 100 Bq/m^3^	25%	16%	7%
Minimum concentration (Bq/m^3^)	14	21	10
Maximum concentration (Bq/m^3^)	1393	598	149
SAA ^1^ average for Sample	77	88	36

Note: ^1^—Seasonal Adjusted Average (SAA).

**Table 2 ijerph-17-04149-t002:** Radon distribution analysis comparing the data in this study against the NRS 2015.

Study	No of Samples	Mean Ratio	Median Ratio	Standard Deviation
National Radon Study	344	0.79	0.74	0.37
Passive House Study	97	1.03	0.92	0.56

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
