# Peer review of "An Investigation into Indoor Radon Concentrations in Certified Passive House Homes"

_ijerph, 2020, doi:10.3390/ijerph17114149_

Round 1

Reviewer 1 Report

The presented paper is an interesting study which try to find some correlation between radon concentration and the type of house (here: passive houses). Generally, the paper is worth to publish, however, I see many problems which need to be consider prior to publication.

General comment: the manuscript needs to be strongly rebuild up to typical research articles' scheme, where methods and results are clearly indicated (it is not necessary to add "Methods" as a section, but this approach should be naturally occuring). At this moment, the manusript seems to be more popular science article than full research paper. The presentation of arguments and results (including Figures) need to be strongly enhanced.

1) Abstract: this abstract in non-informative, 1/3 of it can be simply deleted. Please clearly state: a) what was measured, b) what for, c) methods, d) what are the results and conclusions (brefly of course)

2) Results presentation: the paper suffers for very poor results presentation. Figures need to be corrected (see next items). Results need to be presented in a correct scientific manner including uncertainties (which are completely missing), statistical methods, significance of results, differences in passive houses etc.

3) It should be more detaily explained why passive houses were taken into consideration for this study. What was the scientific hypothesis for that? There are some information for that but, in my opinion, it is not sufficient.

3a) Authors wrote:
"The results support the hypothesis that certified Passive House buildings perform better in respect to indoor radon concentrations compared to conventional homes given less airtightness and no MVHR systems"

Is it your hypothesis (if so, please add that statement) or not (if so, please add the reference).

3b) "The clearest illustration of this was in the presentation of the ten case study dwellings in high risk areas, which highlighted elevated levels in the standard homes compared with the ten Passive House buildings"

What presentation? There is no reference. Additionally, 10 samples study is not a study: the statistics of that measurement is absolutely not sufficient. Especially for radon measuremets where fluctuations can be very large even on the small area.

4) Figures need higher quality, some of them are hard to read (e.g. Fig. 1). Additionally, all figures need to enhance their descriptions / legends.

5) Fig. 2: please explain AL and TL.

6) Fig. 3 and 4: it is hard to find what this picture presents, how many samples, measurements etc.

7) Fig. 5: this picture has completely no description, no information about axes, what case studies, what does it mean "standard" and "PH" etc.

8) Authors wrote:
"The WHO has identified radon as a known human carcinogen and has reported a wealth of biological and epidemiological evidence connecting radon exposure and lung cancer [17]."

"Radon is one of the most dangerous contaminants within the IAQ spectrum. The accumulation in houses can increase the risk of lung cancer (...)."

This is just a sample of wider discussion about the radon influence to human health. Some studies presented that radon causes lung cancer, some studies showed something opposite (see: radon spa), some studies (most of them) showed nothing statistically significant among low radon concentrations. Therefore standard radiation protection approach was applied here: the data from high doses (concentrations) were extrapolated to low doses using ALARA principle, and conservatively mentioned that there is always some risk. After that all regulators or commissions (WHO, IAEA, EC etc.) applied this approach.

In my opinion all scientific papers which describe radon risk (including Authors' manuscript) should add just few words that:
- there are many scientific studies which show radon risk, but there are many studies which show no risk at all - and they cannot be omitted
- the conservative approach to radon risk (where we assume, just for safety, that all concentrations are dangerous) is purely regulation based because scientific background for that was not confirmed in all studies
- I suggest to the Authors to read some papers about that problem, e.g.
* Scott, B.R. Residential radon appears to prevent lung cancer. Dose-Response, 9: 444-464; 2011
* Scott, B.R. Epidemiologic studies cannot reveal the true shape of the Dose-Response relationship for radon-induced lung cancer. Dose-Response, January-March 2019: 1-5.
* Thompson, R.E. Epidemiological evidence for possible radiation hormesis from radon exposure: a case-control study conducted in Worcester, MA. Dose-Response, 2011;9(1):59-75
* Thompson, R.E.; Nelson, D.F.; Popkin, J.H.; Popkin, Z. Case-control study of lung cancer risk from residential radon exposure in Worcester County, Massachussetts. Health Phys, 94: 228-241; 2008
* Dobrzyński, L.; Fornalski, K.W.; Reszczyńska, J. Meta-analysis of thirty two case-control and two ecological radon studies of lung cancer. J Radiat Res, 59(2): 149-163; 2018
* Becker, K. One century of radon therapy. Int. J. Low Radiation, 2004, Vol. 1, No. 3
etc.

9) Authors wrote:
"Radon is estimated to cause 1,100 deaths per year in the UK (and is the second largest identified cause of lung cancer after smoking) [18] and approximately 300 cases of lung cancer in Ireland every year can be linked to radon [19]."

As Authors wrote, it is an estimation. It is not a real clinical study, and this sentence needs to be added. This type of estimations have usually many problems from the statistical point of view, and there are some studies which point that, e.g.: K.W. Fornalski, R. Adams, W. Allison, et al: The assumption of radon-induced cancer risk. Cancer Causes & Control, vol. 26, no. 10, 2015, 1517-1518

Reviewer 2 Report

The investigation of the radon concentration in passive houses is a very important topic because the number of this house increased in the last years. Because of the special ventilation system, the indoor radon level is influenced strongly.

I have some comments/question for the manuscript:

  1. The quality of the figures is very poor. Please change them.
  2. In the introduction, you wrote, that it is the first study in the passive houses about the radon. Is it the first in the world, or do you know other studies?
  3. There are some errors in the text. The "type-correction" is necessary. 

Reviewer 3 Report

Dear authors,

I have read your manuscript and I found it in an adequate form to bo published as it is. Therefor I reccomend it for publications in its present form.

Reviewer 4 Report

A concept of the paper is really interesting for me. However, the structure of the manuscript is very complicated. So I would like to recommend that the authors should improve the manuscript according to the following comments and then re-submit your paper as new submission. Additionally, the authors should read an instruction carefully.

1) The affiliation number of the second author is not 2 but 1. His affiliation is same with the first author, so, please do not repeat same affiliation. The e-mail address of the second author appear in the next of e-mail address of the first author.

2) I do not find subsection 1.1 in the introduction. But the author described 1.2 to 1.6. I guess, introduction is finished before 1.2 in the second page. However, I do not understand the necessity from subsection 1.2 to 1.6. This is not review paper but original article. Thus, if the authors want to explain something about the situation in your country, please include these facts in the Introduction section as a background of the presented study. Usually, introduction section is not divided into several subsection. Especially, radon monitoring described repeatedly. Please explain briefly what is definition and values of the TL and AL. There are one of the key values in this study. Additionally, many countries introduced a reference level for indoor radon according to the IAEA BSS. Of course, I understand that we can find these values after the next page.  

3) Please find the related papers for discuss your results deeply. In fact, the authors cited only four scientific papers. Additionally, references should listed according to the instruction.

4) Please make a clear figure. I do not want to recommend to use the Excel graph without adjustment of size, line and so on. Especially, the authors did not indicate the X and Y axises in Figure 2 and 4. 

Round 2

Reviewer 1 Report

no comments

Author Response

Response to Reviewer 1 Comments

The presented paper is an interesting study which try to find some correlation between radon concentration and the type of house (here: passive houses). Generally, the paper is worth to publish, however, I see many problems which need to be consider prior to publication.

General comment: the manuscript needs to be strongly rebuild up to typical research articles' scheme, where methods and results are clearly indicated (it is not necessary to add "Methods" as a section, but this approach should be naturally occuring). At this moment, the manusript seems to be more popular science article than full research paper. The presentation of arguments and results (including Figures) need to be strongly enhanced.

Point 1: Abstract: this abstract in non-informative, 1/3 of it can be simply deleted. Please clearly state a) what was measured, b) what for, c) methods, d) what are the results and conclusions (briefly of course)

Response 1: The abstract has been re written to reflect these comments. 

Point 2: Results presentation: the paper suffers for very poor results presentation. Figures need to be corrected (see next items). Results need to be presented in a correct scientific manner including uncertainties (which are completely missing), statistical methods, significance of results, differences in passive houses etc.

Response 2: The constructive criticism is most welcome, significant effort has gone into improving the figures and we have now included the uncertainties and elaborated on the significance of the results.

Point 3: It should be more detail explained why passive houses were taken into consideration for this study. What was the scientific hypothesis for that? There are some information for that but, in my opinion, it is not sufficient.

3a) Authors wrote: 
"The results support the hypothesis that certified Passive House buildings perform better in respect to indoor radon concentrations compared to conventional homes given less airtightness and no MVHR systems"

Is it your hypothesis (if so, please add that statement) or not (if so, please add the reference).

3b) "The clearest illustration of this was in the presentation of the ten case study dwellings in high risk areas, which highlighted elevated levels in the standard homes compared with the ten Passive House buildings"

What presentation? There is no reference. Additionally, 10 samples study is not a study: the statistics of that measurement is absolutely not sufficient. Especially for radon measuremets where fluctuations can be very large even on the small area.

Response 3: These comments have been taken on board and again we have now clearly stated the hypotheses ie (3a) and in relation to (3b) the we accepted the commnets the section is now more clear and contextualised within the paper, we also added correctly the point about variance and proximity.

Point 4: Figures need higher quality, some of them are hard to read (e.g. Fig. 1). Additionally, all figures need to enhance their descriptions / legends.

Response 4: Figure 1 is now replaced with a diagram more directly pertinent to the context.

Point 5: Fig. 2: please explain AL and TL.

Response 5: The graph in figure 2 has now been changed to reflect the general comments and the error bars are now also incorporated to capture the uncertainties. In addition, the AL and TL has been addressed in the text and in the caption.

Point 6: Fig. 3 and 4: it is hard to find what this picture presents, how many samples, measurements etc.

Response 6: Figures 3 and 4 have been had graphs redone and have also included the sample size for each subset.

Point 7: Fig. 5: this picture has completely no description, no information about axes, what case studies, what does it mean "standard" and "PH" etc.

Response 7: The graph is figure 5 has been completely changed to reflect the comments.

Point 8: Authors wrote: "The WHO has identified radon as a known human carcinogen and has reported a wealth of biological and epidemiological evidence connecting radon exposure and lung cancer [17]."

"Radon is one of the most dangerous contaminants within the IAQ spectrum. The accumulation in houses can increase the risk of lung cancer (...)."

This is just a sample of wider discussion about the radon influence to human health. Some studies presented that radon causes lung cancer, some studies showed something opposite (see: radon spa), some studies (most of them) showed nothing statistically significant among low radon concentrations. Therefore standard radiation protection approach was applied here: the data from high doses (concentrations) were extrapolated to low doses using ALARA principle, and conservatively mentioned that there is always some risk. After that all regulators or commissions (WHO, IAEA, EC etc.) applied this approach.

In my opinion all scientific papers which describe radon risk (including Authors' manuscript) should add just few words that:
- there are many scientific studies which show radon risk, but there are many studies which show no risk at all - and they cannot be omitted
- the conservative approach to radon risk (where we assume, just for safety, that all concentrations are dangerous) is purely regulation based because scientific background for that was not confirmed in all studies

- I suggest to the Authors to read some papers about that problem, e.g.
* Scott, B.R. Residential radon appears to prevent lung cancer. Dose-Response, 9: 444-464; 2011
* Scott, B.R. Epidemiologic studies cannot reveal the true shape of the Dose-Response relationship for radon-induced lung cancer. Dose-Response, January-March 2019: 1-5. 
* Thompson, R.E. Epidemiological evidence for possible radiation hormesis from radon exposure: a case-control study conducted in Worcester, MA. Dose-Response, 2011;9(1):59-75
* Thompson, R.E.; Nelson, D.F.; Popkin, J.H.; Popkin, Z. Case-control study of lung cancer risk from residential radon exposure in Worcester County, Massachussetts. Health Phys, 94: 228-241; 2008
* Dobrzyński, L.; Fornalski, K.W.; Reszczyńska, J. Meta-analysis of thirty two case-control and two ecological radon studies of lung cancer. J Radiat Res, 59(2): 149-163; 2018
* Becker, K. One century of radon therapy. Int. J. Low Radiation, 2004, Vol. 1, No. 3 
etc.

Response 8: On reflection these comments are accepted, and the recommended reading was interesting and stimulating a number of these papers have now been included (both from Scott) and this narrative has been added to the text to reflect a better balance. The sections highlighted have been removed or reworded.

Point 9: Authors wrote: "Radon is estimated to cause 1,100 deaths per year in the UK (and is the second largest identified cause of lung cancer after smoking) [18] and approximately 300 cases of lung cancer in Ireland every year can be linked to radon [19]."

As Authors wrote, it is an estimation. It is not a real clinical study, and this sentence needs to be added. This type of estimations have usually many problems from the statistical point of view, and there are some studies which point that, e.g.: K.W. Fornalski, R. Adams, W. Allison, et al: The assumption of radon-induced cancer risk. Cancer Causes & Control, vol. 26, no. 10, 2015, 1517-1518

Response 9: Again, like the previous comment (8) the recommended reading was considered and effort was made to reflect the balance in the text.

Response to Reviewer 1 Comments Round 2

Open Review

English language and style

( ) Extensive editing of English language and style required 
( ) Moderate English changes required 
( ) English language and style are fine/minor spell check required 
(x) I don't feel qualified to judge about the English language and style 

Yes

Can be improved

Must be improved

Not applicable

Does the introduction provide sufficient background and include all relevant references?

( )

(x)

( )

( )

Is the research design appropriate?

( )

(x)

( )

( )

Are the methods adequately described?

( )

(x)

( )

( )

Are the results clearly presented?

( )

(x)

( )

( )

Are the conclusions supported by the results?

( )

(x)

( )

( )

Comments and Suggestions for Authors

no comments

Submission Date

30 April 2020

Date of this review

24 May 2020 19:22:06

The reviewer didn’t leave specific comments in round 2, however reflecting on the previous comments and taking into account the matrix above further efforts have been made to address issues in the article to correspond with the areas of improvement indicated above.

Reviewer 4 Report

  1. The number of Dr. Shane Colclough is NOT 2 but 3.
  2. As I mentioned previous version, there is a big problem with the structure of the manuscript. Especially, in introduction. I understood that the authors try to modify the manuscript. However, it is still not enough. In general, "Introduction" should provide background information to support the motivation for the study, and state the study objectives or hypotheses. Additionally, it should needs briefly review the literature to summarize current knowledge. Furthermore, it should be identified the knowledge gaps addressed by the current study.
  3. "Results, Discussion" should be "Results and Discussion"
  4. Please delete the title in Figure 2, 3, 4, and 5. That is, "Certified Passive House Radon Monitoring Results" in Figure 2, "Radon Distribution Average Comparison" in Figure 3, "Construction Materials Radon Comparison" in Figure 4 and "Direct Comparison - Case Studies" in Figure 5.
  5. Please delete "Barry McCarron would like to acknowledge the contribution from Dr. Xianhai Meng and Dr. Shane Colclough with the development of this paper." from the acknowledgments because there have contribution of this manuscript as co-authors.

Author Response

Response to Reviewer 4 Comments

A concept of the paper is really interesting for me. However, the structure of the manuscript is very complicated. So I would like to recommend that the authors should improve the manuscript according to the following comments and then re-submit your paper as new submission. Additionally, the authors should read an instruction carefully.

The affiliation number of the second author is not 2 but 1. His affiliation is same with the first author, so, please do not repeat same affiliation. The e-mail address of the second author appear in the next of e-mail address of the first author.

Response 1: The affiliations have been corrected of the first author has changed affiliation on account of the organisation which are paying the Article Processing Charge (APC).

Point 2: I do not find subsection 1.1 in the introduction. But the author described 1.2 to 1.6. I guess, introduction is finished before 1.2 in the second page. However, I do not understand the necessity from subsection 1.2 to 1.6. This is not review paper but original article. Thus, if the authors want to explain something about the situation in your country, please include these facts in the Introduction section as a background of the presented study. Usually, introduction section is not divided into several subsection. Especially, radon monitoring described repeatedly. Please explain briefly what is definition and values of the TL and AL. There are one of the key values in this study. Additionally, many countries introduced a reference level for indoor radon according to the IAEA BSS. Of course, I understand that we can find these values after the next page.  

Response 2: This commentary is fair, the subsections have been removed and the content has been condensed in the introduction. The point about radon monitoring has been addressed, the final point on the TL and the AL was also address in the new submission.

Point 3: Please find the related papers for discuss your results deeply. In fact, the authors cited only four scientific papers. Additionally, references should list according to the instruction.

Response 3: Effort has been made to discuss the results in more detail and the more scientific papers have been included in the referencing in the context. The References have also been redone to reflect the instruction.  

Point 4: Please make a clear figure. I do not want to recommend using the Excel graph without adjustment of size, line and so on. Especially, the authors did not indicate the X and Y axises in Figure 2 and 4. 

Response 4: All figures in the paper have be redone.

Response to Reviewer 4 Comments Round 2

Point 1: The number of Dr. Shane Colclough is NOT 2 but 3.

Response 1: This has been changed as directed

Point 2: As I mentioned previous version, there is a big problem with the structure of the manuscript. Especially, in introduction. I understood that the authors try to modify the manuscript. However, it is still not enough. In general, "Introduction" should provide background information to support the motivation for the study, and state the study objectives or hypotheses. Additionally, it should needs briefly review the literature to summarize current knowledge. Furthermore, it should be identified the knowledge gaps addressed by the current study.

Response 2: In relation to the commentary above significant effort has been made now in the introduction and with the paper overall to address the concerns of the reviewer. We feel these issues has now been addressed.

Point 3: "Results, Discussion" should be "Results and Discussion"

Response 3: This has been changed as directed and with the new discussion section it is also no longer applicable

Point 4: Please delete the title in Figure 2, 3, 4, and 5. That is, "Certified Passive House Radon Monitoring Results" in Figure 2, "Radon Distribution Average Comparison" in Figure 3, "Construction Materials Radon Comparison" in Figure 4 and "Direct Comparison - Case Studies" in Figure 5.

Response 4: The corresponding graph titles have all been amended as directed above for figures 2,3,4,5, and the new figure 6 which was added in this revision.

Point 5: Please delete "Barry McCarron would like to acknowledge the contribution from Dr. Xianhai Meng and Dr. Shane Colclough with the development of this paper." from the acknowledgments because there have contribution of this manuscript as co-authors.

Response 5: Thank you for spotting this and it has been changed accordingly.